# A New Species of the Genus *Takydromus* (Squamata: Lacertidae) from Northeastern Guangxi, China [note 1]

**DOI:** 10.3390/ani14101402

**Published:** 2024-05-07

**Authors:** Kun Guo, Yong-Hao Hu, Jian Chen, Jun Zhong, Xiang Ji

**Affiliations:** 1Zhejiang Provincial Key Laboratory for Water Environment and Marine Biological Resources Protection, College of Life and Environmental Sciences, Wenzhou University, Wenzhou 325035, China; guokun8808@wzu.edu.cn (K.G.); 21211270108@stu.wzu.edu.cn (Y.-H.H.); chenjian20233@163.com (J.C.); zhongjun@wzu.edu.cn (J.Z.); 2Institute for Eco-Environmental Research of Sanyang Wetland, Wenzhou University, Wenzhou 325014, China

**Keywords:** Lacertidae, molecular phylogenetic analysis, morphology, South China, *Takydromus guilinensis* sp. nov., taxonomy

## Abstract

**Simple Summary:**

The genus *Takydromus* (Squamata: Lacertidae) currently comprises 24 species distributed in East (Eastern Palearctic) and Southeast (Oriental) Asia. Of these 24 species, 15 can be found in China, 8 only in the Chinese mainland and adjacent countries or regions, 6 only in Taiwan and 1 (*T. kuehnei*) on both sides of the Taiwan Strait. Here, we described a new *Takydromus* species from the suburb of Guilin, northeastern Guangxi, South China, based on morphological and mitochondrial DNA data. From the phylogeny reconstructed with a mitochondrial DNA fragment (*CO1* and cyt *b*), we know that the new species differs from its congeners and that it is a sister taxon to *T. intermedius*. Morphologically, the new species can be diagnosed from other *Takydromus* species from the same clade. Based on the above multiple lines of evidence, we suggest that this lacertid lizard from Guilin should be named as a new species, *Takydromus guilinensis* sp. nov. The discovery of this species increases the total number of *Takydromus* species to 25, of which 16 can be found in China.

**Abstract:**

During our collecting trip to Guangxi in 2016, we collected ten specimens of the genus *Takydromus* from the suburb of Guilin, northeastern Guangxi, South China, and found that they did not belong to any currently known species. Here, we described this new species, *Takydromus guilinensis* sp. nov., based on morphological and mitochondrial DNA (*CO1* and cyt *b*) data. This new species is a sister taxon to *T. intermedius* with a *p*-distance of 0.070 in *CO1* and 0.080 in cyt *b*. These two *p*-distances exceed not only the minimum value (0.067) between *T. septentrionalis* and *T. stejnegeri* but also the minimum value (0.079) between *T. intermedius* and *T. yunkaiensis*. Morphologically, this new species differs from other currently recognized *Takydromus* species from the same clade, more evidently in the longitudinal rows of dorsal scales, transverse rows of scales at the mid-body and mensural variables. The description of *Takydromus guilinensis* sp. nov. increases the total number of *Takydromus* species to 25, of which 16 can be found in China. *Takydromus guilinensis* sp. nov. is currently known only from Guilin, Guangxi, South China, where it is sympatric with the other four *Takydromus* species (*T. septentrionalis*, *T. kuehnei*, *T. sexlineatus* and *T. intermedius*).

## 1. Introduction

Asian grass lizards of the genus *Takydromus* Daudin, 1802 (family Lacertidae) are small, oviparous, diurnal species that are distributed in East (Eastern Palearctic) and Southeast (Oriental) Asia; southwestward to Northeastern India; southward to Borneo, Malaya, Sumatra, Bangka and Java; and northward to the Russian Far East [1,2,3]. The degree of local endemism is high in the genus, with 75% of species restricted to only one country [4]. Of the 24 currently recognized *Takydromus* species [4], 6 have been recently recorded since 2001, 4 (*T. albomaculosus* [5] and *T. yunkaiensis* [6] from the Chinese mainland and *T. luyeanus* and *T. viridipunctatus* [7] from Taiwan) from China and 2 (*T. hani* [8] and *T. madaensis* [9]) from Vietnam. There are currently 15 species of *Takydromus* lizards in China, 8 only in the Chinese mainland and adjacent countries or regions (e.g., *T. amurensis* and *T. wolteri* in Far East Russia and Korea and *T. sexlineatus* in Southeast Asia and India), 6 only in Taiwan and 1 (*T. kuehnei*) on both sides of the Taiwan Strait [4].

Four *Takydromus* species have been recorded from the Guangxi Zhuang Autonomous Region (provincial level), South China, including *T. intermedius*, *T. kuehnei*, *T. septentrionalis* and *T. sexlineatus* [10]. During our collecting trip to Guangxi in 2016, we collected 10 specimens of the genus *Takydromus* from the suburb of Guilin (25.3° N, 110.3° E; Figure 1), the second largest city in Guangxi, and found that they did not belong to any currently known species. Careful examination of these specimens, as well as morphological and phylogenetic comparisons between the undescribed lizard and other currently recognized *Takydromus* species, has led to the discovery of considerable differences between them. We therefore suggest that this *Takydromus* lizard should be named as a new species.

## 2. Materials and Methods

### 2.1. Animal Collection and Care

We collected 10 type specimens [5 adult females and 5 adult males larger than 39.4 mm snout–vent length (SVL)] of the new species in early April 2016 from the suburb of Guilin (25.3° N, 110.3° E; Figure 1), Guangxi. Living type specimens were brought to our laboratory, where they were housed together in a 500 × 300 × 200 mm (length × width × height) plastic cage placed in a room where temperatures varied from 20 °C to 28 °C. Thermoregulatory opportunities were provided between 07:00 and 19:00 h by a 60 W full-spectrum lamp suspended over one end of the cage. Mealworms (larvae of *Tenebrio molitor*), house crickets (*Acheta domestica*) and water enriched with multivitamins and minerals were provided daily. Two females laid eggs, one in late April and one in early May. We weighed and measured the eggs and postpartum females less than 6 h post-laying and then incubated the eggs in an incubator (Binder, Tuttlingen, Germany) at 28 °C until hatching. We collected and weighed the hatchlings less than 6 h post-hatching. We deposited all type specimens of the new species in the Research Center of Herpetology at Wenzhou University.

### 2.2. Collection of Mensural and Meristic Data

We followed Wang et al. [6] to collect the mensural and meristic data from each of the 10 specimens, thereby making our data more comparable to those reported previously for *Takydromus* species. The mensural measurements taken for each of the 10 specimens with Mitutoyo digital calipers included the SVL, head length (from the snout to the anterior edge of the tympanum), head width (taken at the posterior end of the mandible), head height (the deepest point on the head), skull length (from the snout to the posterior edge of the occipital), snout–eye length (from the snout to the anterior edge of the eye), abdomen length (the distance between the points of insertion of the fore- and hind-limbs), snout–arm length (from the snout to the anterior edge of the forelimb), ulna length, hind-limb length (femur plus tibia), tibia length and 4th toe length (Appendix A). The meristic data collected for each specimen included the chin-shields, femoral pores, supralabials, infralabials, supraoculars, supraciliary and supratemporal scales, anterior dorsal scale rows (counted transversely at the position of the forelimbs), posterior dorsal scale rows (counted transversely at the position of the hind-limbs), transverse rows of the dorsal scales, longitudinal rows of the dorsal scales, transverse dorsal scale rows at the mid-body, small flat and granular scales in a transverse row on the flank at the mid-body (SSRF), transverse rows of the ventral scales, longitudinal rows of the ventral scales, enlarged and keeled scale rows above the ventral scales on the flank (ESRF), caudal scale rows (counted around the tail in the position of the 11th to 13th subcaudal scales, CSR), subdigital lamellae under the 4th finger and subdigital lamellae under the 4th toe (Appendix A).

### 2.3. DNA Isolation and Sequencing 

We used the DNeasy tissue kit (Qiagen, Germany) to extract the total genomic DNA from each of the five tail muscle samples (three from females and two from males) according to the manufacturer’s instructions. We used the primers *CO1* R: 5′-GTG CCT GAG CTG GCA TAG TT-3′/*CO1* F: 5′-GTT CCG CCA TGA AGT GTT GC-3′ and cyt *b* R: 5′-ATT AAC GCT TCT TTT ATT GAC CTA CC-3′/cyt *b* F: 5′-AAG AAA AAT TAT AAA GTA AAG GGC TG-3′ to amplify and sequence the cytochrome oxidase 1 (*CO1*) and cytochrome *b* (cyt *b*) gene fragments, respectively. Polymerase chain reaction (PCR) was performed in 30 μL reaction volumes and under the following cycling conditions: 5 min at 94 °C, followed by 35 cycles of 45 s at 94 °C, 1 min at the annealing temperature (55 °C for *CO1*, and 56 °C for cyt *b*), 90 s at 72 °C and 10 min at 72 °C. The amplifications were performed with Taq DNA polymerase (TaKaRa, Dalian, China) and on the PCR Instrument (Biometra Tone 96, Jena, Germany). The products were then forward- and reverse-sequenced to ensure the accuracy of the result by Invitrogen Biotechnology (Shanghai, China).

### 2.4. Data Analyses and Phylogeny

We used principal components analysis (PCA) to show the positions of the undescribed species and other known *Takydromus* species from the same clade found in China on a two-dimensional plane defined by 12 mensural morphological variables not including the body mass and tail length (Appendix A), using one-way analysis of variance (ANOVA) to examine whether the mean factor scores of the first two PC axes differed among species from the same clade. We used Z-scores to standardize the mensural morphological data, thereby avoiding the confounding influence of dimensional differences among variables. All the statistical analyses were performed with Statistica 14.0 (Tulsa, OK, USA), and the significance level was set at α = 0.05.

The sequence data were assembled and aligned using Seqman and MegAlign in Lasergene [11]. The mitochondrial sequencing data from the other 17 known *Takydromus* species and *Eremias vermiculata* were obtained from GenBank (NCBI, Bethesda, MD, USA; Table 1). We estimated the best partition scheme and evolutionary model for the concatenation dataset of the *CO1* and cyt *b* gene sequences with PartitionFinder 2 to analyze the phylogenetics of the mitochondrial DNA [12]. After that, we used IQ-TREE [13] under the best-fit GTR+I+G4+F model for 5000 ultrafast bootstraps approximation to construct a Maximum Likelihood (ML) tree and MrBayes 3.2.6 [14] under the best-fit model GTR+I+G+F (2,000,000 generations) with the initial 25% of the sampled data discarded as burn-in to construct the Bayesian Inference (BI) tree in PhyloSuite 1.2.2 [15].

## 3. Results

We obtained a concatenated mtDNA alignment of 1418 bp (402 bp *CO1* and 1016 bp cyt *b*) from each of the five samples. The ML and BI phylogenetic trees yielded essentially identical topologies and were therefore combined into one with high bootstrap support values (BS ≥ 70) and Bayesian posterior probabilities (BPP ≥ 0.65) (Figure 2). *Takydromus guilinensis* sp. nov. and *T. intermedius* were clustered in the same clade with strong supports (BPP = 1.00 and BS = 93) but showed distinct divergence in the genes *CO1* (*p*-distance = 0.070) and cyt *b* (*p*-distance = 0.080) (Table 2).

The longitudinal rows of the dorsal scales and transverse dorsal scale rows at the mid-body in the *T. guilinensis* sp. nov. specimens were 32–36 and 27–33, respectively. These meristic morphological variables differed from those recorded in other known *Takydromus* species from the same clade (Table 3). The PCA of the 12 mensural morphological variables revealed that the mean scores on the first (*F*_4,24_ = 10.40, *p* < 0.01) and second (*F*_4,24_ = 18.21, *p* < 0.001) axes differed significantly among the five *Takydromus* species from the same clade, including *T. guilinensis* sp. nov. (Appendix A, Figure 3). Overall, the morphological differences between *T. guilinensis* sp. nov. and the other four *Takydromus* species from the same clade were evident (Figure 3).

Two female *T. guilinensis* sp. nov. laid a clutch of two eggs each, one in late April and one in early May. The egg masses ranged from 154.9 mg to 217.8 mg, with a mean of 186.9 mg.

Two eggs, one in each clutch, hatched, with the incubation length at 28 °C being 37 d in one egg and 33 d in the other egg. The hatchling masses ranged from 195.7 mg to 244.8 mg, with a mean of 220.3 mg (Table 4).

Taxonomy*Takydromus guilinensis* sp. nov. (Figure 4A–F)

Holotype. WZU R20160406, adult female, from the suburb of Guilin (25.3° N, 110.3° E; Figure 1A), Guangxi, China, collected in early April 2016 by Kun Guo.Paratype. WZU20160401R, adult male, from the suburb of Guilin, Guangxi, China, collected in early April 2016 by Kun Guo.Etymology. The specific epithet is a Latinization of Guilin City, Guangxi, China.Diagnosis. A new small oviparous *Takydromus* species distinguished from all other currently known congeneric species with the following mensural and meristic characters. First, SVL 39.4–46.7 mm in adult males and 42.8–52.4 mm in adult females. Second, dorsal ground color brown; ventral surface white. Third, skull flattening (HL/HW = 1.8–2.0). Fourth, 32–36 longitudinal rows of dorsal scales. Fifth, 27–33 transverse rows of scales at the mid-body.Description of Holotype. Adult female. SVL 52.4 mm, tail length 150.1 mm, head length 11.8 mm, head width 6.7 mm, head height 4.6 mm, skull length 12.4 mm, snout–eye length 8.6 mm, arm–leg length 23.8 mm, snout–arm length 18.0 mm, radius–ulna length 5.1 mm, hind-limb length 14.4 mm, tibia–fibula length 6.3 mm, length of the 4th toe on the hind-limb 8.2 mm, chin-shields 4, femoral pores 3/3, supralabials 6, infralabials 6, supraoculars 5, supraciliaries 4, supratemporals 4/5, anterior dorsal scale rows 6, posterior dorsal scale rows 6, transverse rows of the dorsal scales 7, longitudinal rows of the dorsal scales 32, transverse dorsal scale rows at the mid-body 27, SSRF 10/10, transverse rows of the ventral scales 6, longitudinal rows of the ventral scales 22, ESRF 1, CSR 12, subdigital lamellae under the 4th finger 21/23 and subdigital lamellae under the 4th toe 27/27.Color of Holotype in Life. The dorsal surface of the head, body and limbs grey-brown, tail bright brown; the ventral surface of the head and tail white, body white in the paunch and yellow-green in the pereion and limbs brown; lateral surfaces of the body grey-brown, head grey brown and white distinctly separate in the inferior orbit; a pair of longitudinal bright brown dorsolateral stripes beginning at the supraciliary position, running along the supratemporals and outermost dorsal scale rows, posteriorly extending to the anterior part of the tail.Color Variation. In life, the adult males differed in coloration from the holotype in the ventral surface color of the pereion white.Female Reproduction. Two adult females (SVLs 52.4 mm and 42.9 mm) laid a clutch of two eggs each between late May and early June. The postpartum body masses ranged from 1.48 g to 1.10 g, egg masses from 154.9 mg to 217.8 mg and hatchling masses from 195.7 mg to 244.8 mg (Table 4).Natural History. *Takydromus guilinensis* sp. nov. uses habitats with shallow vegetation not fully covering the ground and scattered rocks at the foot of the mountain, more often basks on stone blocks and retreats into shelters such as rock crevices after being disturbed. No other *Takydromus* lizards in Guangxi use the same habitats as does *Takydromus guilinensis* sp. nov.

## 4. Discussion

*Takydromus guilinensis* sp. nov. is sympatric with the other four *Takydromus* lizards (*T. septentrionalis*, *T. kuehnei*, *T. sexlineatus* and *T. intermedius*) in Guangxi, where the species richness of the genus *Takydromus* is only lower than in Taiwan (7 species) and Guangdong (6 species) Provinces [4]. The new species is a sister taxon to *T. intermedius*, with a significant divergence of uncorrected *p*-distances of 0.070 in *CO1* and 0.080 in cyt *b* (Table 2). These two *p*-distances exceed not only the minimum value (0.067) between *T. septentrionalis* and *T. stejnegeri* but also the minimum value (0.079) between *T. intermedius* and *T. yunkaiensis* (Table 2). Morphologically, *T. guilinensis* sp. nov. differs from the other four *Takydromus* species from the same clade, more evidently in the longitudinal rows of the dorsal scales (up to 36 in *T. guilinensis* but at least 36 in *T. intermedius*, 47 in *T. yunkaiensis*, 52 in *albomaculosus* and 67 in *T. sylvaticus*), transverse rows of scales at the mid-body (up to 33 in *T. guilinensis* but at least 40 in *T. intermedius* and *T. yunkaiensis*, 42 in *T. albomaculosus* and 45 in *T. sylvaticus*) and mensural variables (Table 3; Figure 3).

The interrelationships and evolution of Asian grass lizards, *Takydromus* (Squamata: Lacertidae), are still in dispute and, due to the inconsistencies in the molecular and/or morphological data, different conclusions have been drawn [1,7,16,17,18,19]. For example, our phylogenetic tree of two concatenated mtDNA genes (*CO1* and cyt *b*) differs from the trees based on *CO1* [5] or cyt *b* [6] in the positions of *T. smaragdinus*, *T. kuehnei* and *T. sexlineatus* within the genus *Takydromus* (Figure 2). Phylogenetic analyses and geographical patterns of taxonomic diversity support the ancestral form of *Takydromus* deriving from one of the early divergences of the family Lacertidae in western Eurasia, but no biogeographical hypotheses have hitherto been proposed to expound the divergence within this genus [18,20,21,22]. Thus, the issues associated with the evolution relationship and phylogenetic status of the genus *Takydromus* still need to be addressed in future studies.

In total, 19 of the 24 currently recognized *Takydromus* species have a relatively small range, occurring only in several places of a province or several provinces of a country [4]. For example, of the seven *Takydromus* species in Taiwan, six are endemic to this small island province [4]. It is worth noting that all six newly described *Takydromus* species have a narrow range [5,6,7,8,9], not following the process of speciation by physical or ecological isolation [23]. However, widespread sympatry of the *Takydromus* species does occur in the ranges of these six newly described *Takydromus* species, such as *T. kuehnei*, *T. septentrionalis*, *T. sexlineatus* and *T. sylvaticus* in the range of *T. yunkaiensis* [6]; *T. septentrionalis* and *T. kuehnei* in the range of *T. albomaculosus* [5]; *T. kuehnei* and *T. sexlineatus* in the range of *T. madaensis* [9]; *T. formosanus, T. hsuehshanensis, T. sauteri, T. stejnegeri* and *T. kuehnei* in the range of *T. viridipunctatus* and *T. luyeanus* [7]; and *T. kuehnei* and *T. sexlineatus* in the range of *T. hani* [8]. Two widespread *Takydromus* species, *T. septentrionalis* and *T. sexlineatus*, are sympatric in some areas of South China and bidirectional heterospecific matings frequently occur in the laboratory. From these observations, Guo et al. [24] have proposed a hypothesis that hybridization between sympatric closely related species might lead to introgression and speciation. The hybrid speciation has been viewed as a significant hypothesis for studying species barriers and reproductive isolation [25], which might decipher the indetermination of interrelationships and disproportion diversity in *Takydromus* species.

## 5. Conclusions

We described a new lacertid species of the genus *Takydromus*, *Takydromus guilinensis* sp. nov., from the suburb of Guilin, Northeastern Guangxi, South China, based on morphological and mitochondrial DNA data. The description of this new species increases the total number of *Takydromus* species to 25, of which 16 can be found in China. *Takydromus guilinensis* sp. nov. is currently known only from Guilin, Guangxi, South China, where it is sympatric with other four *Takydromus* species (*T. septentrionalis*, *T. kuehnei*, *T. sexlineatus* and *T. intermedius*). Given the high endemicity of the viviparous Chinese toad-headed lizards [4], local endemism of *T. guilinensis* is not unexpected. Future work could usefully investigate the range of distribution, habitat use, population size, food habits, life-history traits, thermal requirements of the new species and mechanisms it adopts to coexist with other sympatric *Takydromus* species.

## Figures and Tables

**Figure 1 animals-14-01402-f001:**
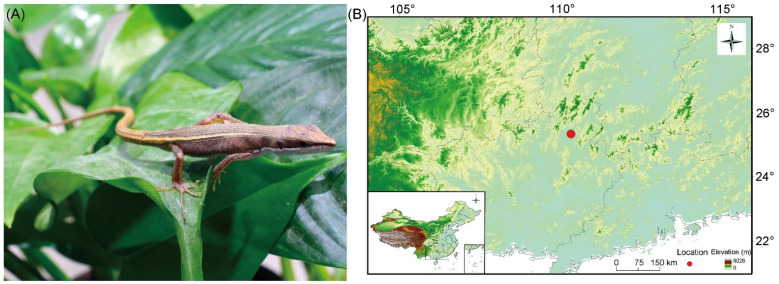
A gravid female *Takydromus guilinensis* sp. nov. (**A**) and the locality where 10 specimens of the new species were collected (**B**).

**Figure 2 animals-14-01402-f002:**
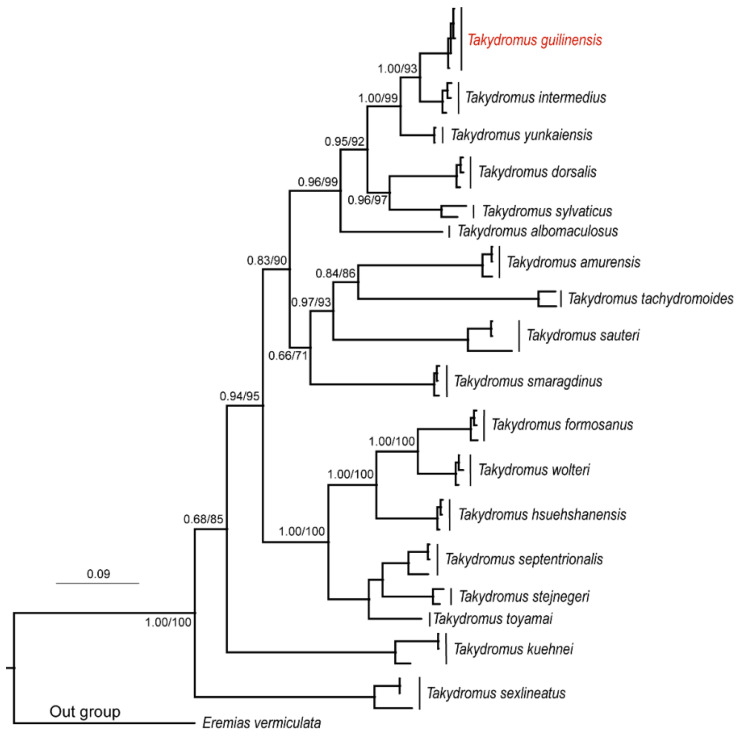
A combined phylogeny obtained by Maximum Likelihood and Bayesian Inference with high bootstrap support values (≥70) and Bayesian posterior probabilities (≥0.65).

**Figure 3 animals-14-01402-f003:**
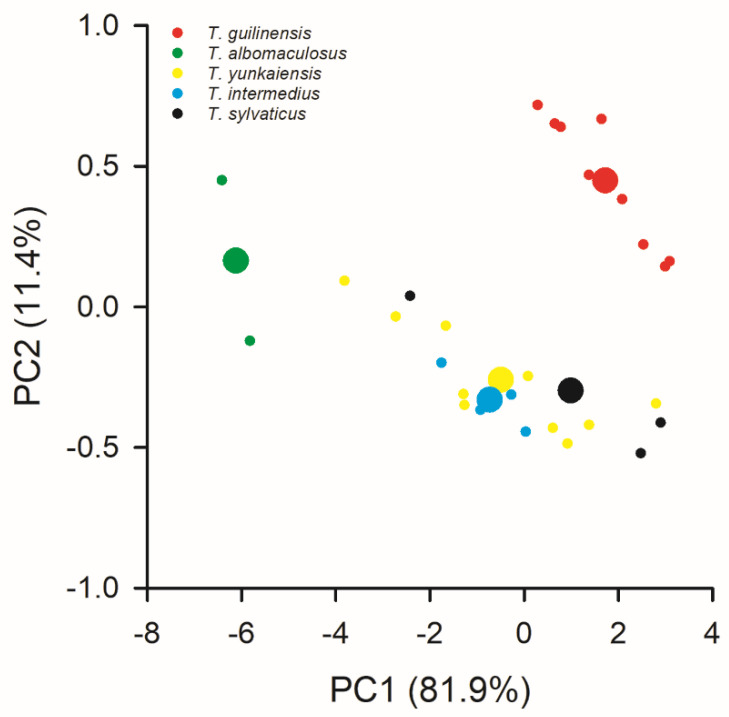
Positions of five *Takydromus* species from the same clade including *Takydromus guilinensis* sp. nov. defined by 12 mensural morphological variables. Effects of body size were removed using residuals from the regressions of corresponding variables on SVL. Larger dots show the mean values of scores on the first two axes.

**Figure 4 animals-14-01402-f004:**
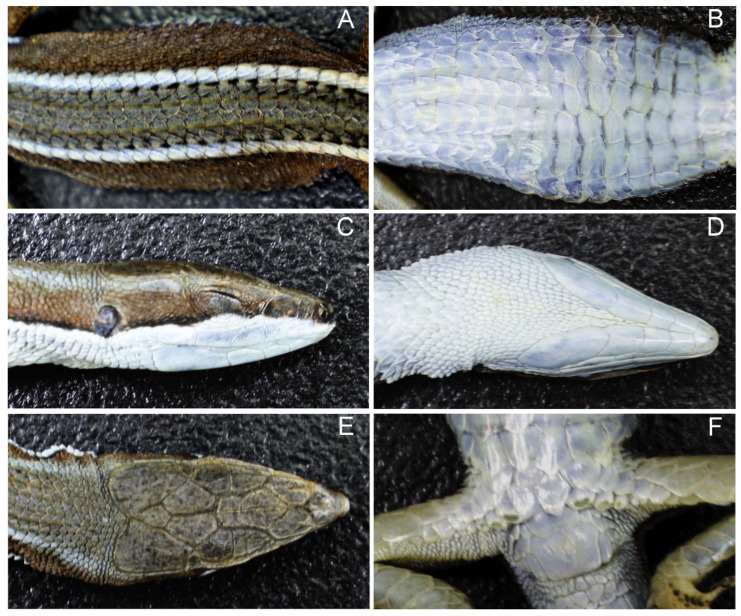
Close-up views of the adult female holotype (WZU R20160406) of *Takydromus guilinensis* sp. nov. from the suburb of Guilin. (**A**): dorsal view of the body; (**B**): ventral view of the body; (**C**): lateral view of the head; (**D**): ventral view of the head; (**E**): dorsal view of the head; (**F**): ventral view of the posterior part of the body, showing inguinal pores. Photo by Kun Guo.

**Table 1 animals-14-01402-t001:** GenBank accession numbers of mitochondrial *CO1* and cyt *b* genes for all sequences used in this study.

ID	Species	GenBank Number	
		*CO1* Gene	Cyt *b* Gene
1	*Takydromus tachydromoides*	AY248555	LC066076
		AY248554	LC066075
2	*Takydromus septentrionalis*	JX196703	JX196702
		AY248545	MN239968
		MK630237	MN239967
3	*Takydromus formosanus*	AY248604	AY248533
		AY248603	AY248532
		AY248602	AY248531
4	*Takydromus hsuehshanensis*	AY248562	AY248486
		AY248561	AY248485
		AY248560	AY248484
5	*Takydromus wolteri*	AY248557	MN239973
		JQ844542	MN239972
		NC_018777	NC_018777
6	*Takydromus toyamai*	AY248556	AY248480
7	*Takydromus stejnegeri*	AY248553	AY248477
		AY248552	AY248476
8	*Takydromus smaragdinus*	AY248551	LC066078
		AY248550	AY248473
		AY248549	AY248474
9	*Takydromus sexlineatus*	NC_022703	MN239970
		KF425529	MN239969
		AY248546	MN015203
10	*Takydromus sauteri*	AY248543	AY248467
		AY248542	AY248466
		AY248541	AY248465
11	*Takydromus kuehnei*	AY248540	MN239964
		AY248539	MN239962
		MZ435950	MN239963
12	*Takydromus intermedius*	AY248538	MN239961
		MF631861	MN239960
		MF631860	MN239959
13	*Takydromus dorsalis*	AY248537	AY248461
		AY248536	AY248460
		—	LC066079
14	*Takydromus amurensis*	AY248535	KU841538
		AY248534	KU841537
		JQ798892	KU841536
15	*Takydromus sylvaticus*	JX290083	EF495176
		MF631871	JX290083
16	*Takydromus albomaculosus*	MF631870	—
17	*Takydromus yunkaiensis*	—	MN239955
		—	MN239954
18	*Takydromus guilinensis*	OR667005	OR820954
		OR667006	OR820955
		OR667007	OR820956
		OR667008	OR820957
		OR667009	OR820958
19	*Eremias vermiculata*	MK261078	MK261078

**Table 2 animals-14-01402-t002:** Mean group distance of *p*-distance between *Takydromus* species based on mitochondrial genes (*CO1*\*Cytb*). *T. formosanus*: T_for; *T. hsuehshanensis*: T_hsu; *T. sexlineatus*: T_sex; *T. septentrionalis*: T_sep; *T. smaragdinus*: T_sma; *T. sauteri*: T_sau; *T. kuehnei*: T_kue; *T. wolteri*: T_wol; *T. amurensis*: T_amu; *T. stejnegeri*: T_ste; *T. dorsalis*: T_dor; *T. tachydromoides*: T_tac; *T. sylvaticus*: T_syl; *T. toyamai*: T_toy; *T. albomaculosus*: T_alb; *T. intermedius*: T_int; *T. yunkaiensis*: T_yun; and *T. guilinensis*: T_gui.

	T_for	T_hsu	T_sex	T_sep	T_sma	T_sau	T_kue	T_wol	T_amu	T_ste	T_dor	T_tac	T_syl	T_toy	T_alb	T_int	T_yun	T_gui
T_for		0.140	0.199	0.158	0.199	0.193	0.198	0.089	0.189	0.156	0.204	0.208	0.238	0.164		0.201	0.211	0.214
T_hsu	0.100		0.201	0.147	0.189	0.189	0.177	0.111	0.203	0.143	0.191	0.208	0.218	0.147		0.182	0.186	0.188
T_sex	0.175	0.148		0.183	0.207	0.201	0.190	0.205	0.215	0.198	0.191	0.202	0.234	0.170		0.187	0.178	0.202
T_sep	0.116	0.104	0.168		0.196	0.195	0.176	0.150	0.180	0.109	0.203	0.200	0.220	0.100		0.187	0.199	0.208
T_sma	0.176	0.152	0.174	0.164		0.171	0.193	0.185	0.173	0.178	0.169	0.170	0.210	0.188		0.164	0.162	0.185
T_sau	0.149	0.143	0.149	0.151	0.159		0.208	0.184	0.189	0.193	0.195	0.209	0.229	0.192		0.195	0.206	0.198
T_kue	0.175	0.171	0.195	0.186	0.175	0.166		0.199	0.205	0.185	0.201	0.199	0.242	0.186		0.202	0.201	0.223
T_wol	0.071	0.109	0.165	0.119	0.168	0.169	0.172		0.175	0.160	0.191	0.209	0.239	0.163		0.208	0.196	0.211
T_amu	0.151	0.156	0.183	0.146	0.168	0.143	0.164	0.149		0.171	0.192	0.182	0.234	0.177		0.171	0.176	0.179
T_ste	0.119	0.111	0.163	0.067	0.178	0.164	0.189	0.136	0.144		0.201	0.207	0.219	0.112		0.199	0.202	0.205
T_dor	0.160	0.165	0.170	0.171	0.153	0.171	0.168	0.152	0.168	0.191		0.168	0.207	0.188		0.129	0.124	0.145
T_tac	0.197	0.191	0.163	0.196	0.174	0.167	0.179	0.185	0.192	0.188	0.163		0.224	0.207		0.182	0.174	0.202
T_syl	0.173	0.160	0.154	0.181	0.176	0.148	0.175	0.177	0.176	0.186	0.113	0.144		0.224		0.204	0.205	0.211
T_toy	0.123	0.104	0.168	0.082	0.174	0.167	0.208	0.131	0.148	0.083	0.172	0.200	0.206			0.187	0.188	0.192
T_alb	0.154	0.163	0.162	0.154	0.135	0.155	0.175	0.155	0.167	0.179	0.126	0.193	0.149	0.184				
T_int	0.145	0.141	0.158	0.157	0.160	0.131	0.167	0.158	0.142	0.164	0.112	0.160	0.113	0.171	0.116		0.079	0.080
T_yun																		0.087
T_gui	0.165	0.167	0.158	0.168	0.160	0.155	0.174	0.157	0.141	0.180	0.104	0.154	0.124	0.168	0.094	0.070		

**Table 3 animals-14-01402-t003:** Meristic variable measured by counting type series of *Takydromus guilinensis* and four other *Takydromus* species from the same clade.

Variable	*T. guilinensis*	*T. yunkaiensis*	*T. albomaculosus*	*T. intermedius*	*T. sylvaticus*
CS	4	4	4	4–5	4
FP	2–3	2–3	3–4	2–3	3
SPL	6	6–7	6–7	6–7	5–7
IFL	5–6	6–7	6–7	5–7	5–7
SPO	4–5	4	3 (rarely 4)	4	4
SPC	4–6	4 (rarely 2, 3)	4–6	4–5	4–5
SPT	2–5	3–4	3	2-5	2-4
ADSR	6	9–10	6	6–8	—
PDSR	6	7	6	6	9–10
MDSR	7	7–8	7	7–8	11–14
LDSN	32–36	47–51	52–53	36–46	67–81
MBSR	27–33	40–46	42–43	40–44	45–47
SSRF	10–14	12–17	13–14	12–15	13
VR	6	6	6	6	6
VN	20–22	24–27	23–26	21–24	26–29
ESRF	1	1	1	1	0
CSR	12	10–13	12	12	12
SDLF-4	20–24	20–23	23–24	20–21	21–22
SDLT-4	24–28	23–30	29–30	26–27	27–28

CS: chin-shields; FP: femoral pores; SPL: supralabials; IFL: infralabials; SPO: supraocular; SPC: supraciliary; SPT: supratemporals; ADSR: anterior dorsal scale rows, distinctly enlarged and keeled scales on the anterior dorsum, counted transversely at the position of the forelimbs; PDSR: posterior dorsal scale rows, counted transversely at the position of the hind-limbs; LDSN: dorsal scale numbers, counted longitudinally from the posterior margin of the occipital to posterior margin of the hind-limbs; MDSR: transverse dorsal scale rows at the mid-body; VR: ventral scale rows, counted transversely at the mid-body; VN: ventral scale numbers, counted longitudinally from the posterior margin of the collars to the anterior margin of the preanal scales; ESRF: enlarged and keeled scale rows above the ventrals on the flank; SSRF: small flat and granular scales in a transverse row on the flank at the mid-body; MBSR: scales in a transverse row at the mid-body, including ventrals; CSR: caudal scale rows, counted around the tail in the position of the 11th to 13th subcaudal scales; and SDLF-4: subdigital lamellae under the fourth finger and fourth toe (SDLT-4).

**Table 4 animals-14-01402-t004:** Information on two clutches laid by female *Takydromus guilinensis* sp. nov. Eggs were incubated at a constant temperature of 28 °C.

# Female	# Egg	Egg Length (mm)	Egg Width (mm)	Egg Mass (mg)	Egg-Laying Date (dd/mm/year)	Hatched Date (dd/mm/year)	Hatchling Mass (mg)
1	01	8.78	5.88	154.9	23 April 2016	30 May 2016	195.7
	02	8.84	5.94	157.5	23 April 2016	—	—
2	03	9.86	6.06	217.3	2 May 2016	—	—
	04	9.84	5.81	217.8	2 May 2016	4 July 2016	244.8

## Data Availability

Data are contained within the article.

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
