# Peer review of "A New Species of the Genus Takydromus (Squamata: Lacertidae) from Northeastern Guangxi, China"

_animals, 2024, doi:10.3390/ani14101402_

Round 1

Reviewer 1 Report

Comments and Suggestions for Authors

The article under review is devoted to the description of a new species of lizards within the Genus Takydromus in China. When characterizing a new species, the authors rely on molecular genetic and morphological characteristics indicating its phylogenetic independence. The criteria used to identify the species are quite justified and are  traditionally used in similar studies. An additional argument in favor of the fact that the  new species does not belong to the allospecies category is its sympatry with other closely related species of the Genus. The authors presented in the article the diagnosis of the new species, its clear diagnostic characteristics, and also assessed phylogenetic relationships with other species of the genus based on the nucleotide sequences of mitochondrial DNA gene fragments. The methods of analysis and study design used do not raise any  objections. The authors presented informative and high-quality illustrations that successfully complement the material presented in the article and facilitate the diagnosis of  the species. T

he results of morphometric analysis performed using the Principal component method are also convincing. All ordinates of individuals of the new species, in comparison with the ordinates of four phylogenetically similar species, are separately localized in the morphospace of the first two Principal components. Of course, it would be ideal to obtain PC ordinates for a fifth closely related species, Takydromus dorsalis, but its  absence in the calculations is not fundamental, since the available results convincingly  prove the morphological remoteness of the new species, Takydromus guilinensis sp. nov.

The design of tables and illustrations does not cause any comments. The text was well checked by the authors, since even after carefully reading it I was unable to identify any typos or errors. I believe that the material in the article is of great scientific importance and will be received with great interest not only by a narrow circle of specialists, but also by a wide range of readers of the Journal. I believe that the paper can be recommended for publication in the Journal of Animals in its presented form without re-reviewing.

Author Response

Reviewer 1

The article under review is devoted to the description of a new species of lizards within the Genus Takydromus in China. When characterizing a new species, the authors rely on molecular genetic and morphological characteristics indicating its phylogenetic independence. The criteria used to identify the species are quite justified and are traditionally used in similar studies. An additional argument in favor of the fact that the new species does not belong to the allospecies category is its sympatry with other closely related species of the Genus. The authors presented in the article the diagnosis of the new species, its clear diagnostic characteristics, and also assessed phylogenetic relationships with other species of the genus based on the nucleotide sequences of mitochondrial DNA gene fragments. The methods of analysis and study design used do not raise any objections. The authors presented informative and high-quality illustrations that successfully complement the material presented in the article and facilitate the diagnosis of the species.

>> Thanks for your positive comments

The results of morphometric analysis performed using the Principal component method are also convincing. All ordinates of individuals of the new species, in comparison with the ordinates of four phylogenetically similar species, are separately localized in the morphospace of the first two Principal components. Of course, it would be ideal to obtain PC ordinates for a fifth closely related species, Takydromus dorsalis, but its  absence in the calculations is not fundamental, since the available results convincingly  prove the morphological remoteness of the new species, Takydromus guilinensis sp. nov.

>> Takydromus dorsalis was not included in PCA because of the lack of comparable data on the species

The design of tables and illustrations does not cause any comments. The text was well checked by the authors, since even after carefully reading it I was unable to identify any typos or errors. I believe that the material in the article is of great scientific importance and will be received with great interest not only by a narrow circle of specialists, but also by a wide range of readers of the Journal. I believe that the paper can be recommended for publication in the Journal of Animals in its presented form without re-reviewing.

>> Thanks for your positive comments

Reviewer 2 Report

Comments and Suggestions for Authors

The presented study is a quality comprehensive description of a new species and a significant expansion of existing knowledge about the taxonomy of the genus Takydromus. I recommend accepting the manuscript for publication. I have only a few formal marginal notes:

I recommend mentioning the name of the author of the Takydromus taxon in the Introduction (the first line)

2.1. Acheta domesticus …. (Acheta is masculine)

The quality of the photos (fig. 4) could be higher

Author Response

Reviewer 2

The presented study is a quality comprehensive description of a new species and a significant expansion of existing knowledge about the taxonomy of the genus Takydromus. I recommend accepting the manuscript for publication. I have only a few formal marginal notes:

>> Thanks for your positive comments

I recommend mentioning the name of the author of the Takydromus taxon in the Introduction (the first line)

>> Done. Thanks

2.1. Acheta domesticus …. (Acheta is masculine)

>> Correct. Thanks

The quality of the photos (fig. 4) could be higher

>> We replaced Figure 4 by a remade one and thereby improved its readability

Reviewer 3 Report

Comments and Suggestions for Authors

The genus of Asian grass lizards Takydromus Daudin, 1802 is highly diverse with at least 24 species and is widely distributed in eastern Asia. A significant part of the species live sympatrically and have clearly distinguishable morphological characteristics. Differences established by taxonomic characteristics require confirmation by molecular genetic analysis. Thus, the presented study is of significant interest.

The manuscript is prepared at a high level, but there are several comments.

There are repetitions of expressions throughout the text, for example:

- 4. Discussion

“...the total number of species of the genus Takydromus to 25.”

- 5.Conclusions

“…the total number of Takydromus species to 25”

- Section “2.2. Collection of Mensural and Meristic Data"

There are no abbreviated designations for metric features, for example in additional materials: “ADSR anterior dorsal scale rows, distinctly enlarged and keeled scales on anterior dorsum, counted transversely at position of forelimbs; CS chin-shields; CSR caudal scale rows, counted around the tail in the position of the 11th to 13th subcaudal scales; ESRF enlarged and keeled lateral scales in longitudinal row(s) above ventrals on lower flanks; FP femoral pores; IFL infralabials; LDSN dorsal scale numbers, counted longitudinally from posterior margin of occipital to posterior margin of hind limbs; MBSR scales in a transverse row at mid-body, including ventrals; MDSR transverse dorsal scale rows at mid-body; PDSR posterior dorsal scale rows, counted transversely at the position of hind limbs; SDLF-IV subdigital lamellae under fourth finger; SDLT-IV subdigital lamellae under fourth toe; SPC supraciliary; SPL supralabials; SPO supraocular; SPT supratemporals; SSRF small flat and granular scales in a transverse row on flank at mid-body; TSRF enlarged and keeled scale rows above ventrals on flank; VN ventral scale numbers, counted longitudinally from the posterior margin of collars to the anterior margin of precloacal scales; VR ventral scale rows, counted transversely at mid-body."

- Missing table files S.1 and S.2

- “Morphologically, this new species differs from other currently recognized Takydromus species from the same clade more evidently in longitudinal rows of dorsal scales, transverse rows of scales at the midbody, and mensural variables.” Further, from the text of the manuscript, differences in 2 characteristics are revealed: LDSN - “dorsal scale numbers, counted longitudinally from posterior margin of occipital to posterior margin of hind limbs” (Takydromus guilinensis up to 36, other species over 36 - T. intermedius, T. yunkaiensis - 47, T. albomaculosus – 52, T. sylvaticus - 67); MBSR - “scales in a transverse row at mid-body, including ventrals” (T. guilinensis up to 33, other species over 40 - T. yunkaiensis and T. intermedius; 42 - T. albomaculosus and 45 - T. sylvaticus).

Design “Figure 1. A gravid female Takydromus guilinensis sp. nov. (A) and the locality where 10 specimens of the new species were collected (B)” requires improvement in part (B).

Add to the discussion the publication “Climate-driven mitochondrial selection in lacertid lizards” (Zhang et al., 2024, doi: 10.1002/ece3.11176).

Author Response

Reviewer 3

The genus of Asian grass lizards Takydromus Daudin, 1802 is highly diverse with at least 24 species and is widely distributed in eastern Asia. A significant part of the species live sympatrically and have clearly distinguishable morphological characteristics. Differences established by taxonomic characteristics require confirmation by molecular genetic analysis. Thus, the presented study is of significant interest. The manuscript is prepared at a high level, but there are several comments.

>> Thanks for your positive comments

There are repetitions of expressions throughout the text, for example:

- 4. Discussion

“...the total number of species of the genus Takydromus to 25.”

- 5.Conclusions

“…the total number of Takydromus species to 25”

>> We deleted “...the total number of species of the genus Takydromus to 25” from the Discussion and kept the sentence in the Conclusion unchanged

- Section “2.2. Collection of Mensural and Meristic Data"

There are no abbreviated designations for metric features, for example in additional materials: “ADSR anterior dorsal scale rows, distinctly enlarged and keeled scales on anterior dorsum, counted transversely at position of forelimbs; CS chin-shields; CSR caudal scale rows, counted around the tail in the position of the 11th to 13th subcaudal scales; ESRF enlarged and keeled lateral scales in longitudinal row(s) above ventrals on lower flanks; FP femoral pores; IFL infralabials; LDSN dorsal scale numbers, counted longitudinally from posterior margin of occipital to posterior margin of hind limbs; MBSR scales in a transverse row at mid-body, including ventrals; MDSR transverse dorsal scale rows at mid-body; PDSR posterior dorsal scale rows, counted transversely at the position of hind limbs; SDLF-IV subdigital lamellae under fourth finger; SDLT-IV subdigital lamellae under fourth toe; SPC supraciliary; SPL supralabials; SPO supraocular; SPT supratemporals; SSRF small flat and granular scales in a transverse row on flank at mid-body; TSRF enlarged and keeled scale rows above ventrals on flank; VN ventral scale numbers, counted longitudinally from the posterior margin of collars to the anterior margin of precloacal scales; VR ventral scale rows, counted transversely at mid-body."

>> Abbreviations for meristic variables are given in Table 3. Only abbreviations for three meristic variables repeatedly mentioned in the text were provided in Section 2.2

- Missing table files S.1 and S.2

>> Added. Thanks

- “Morphologically, this new species differs from other currently recognized Takydromus species from the same clade more evidently in longitudinal rows of dorsal scales, transverse rows of scales at the midbody, and mensural variables.” Further, from the text of the manuscript, differences in 2 characteristics are revealed: LDSN - “dorsal scale numbers, counted longitudinally from posterior margin of occipital to posterior margin of hind limbs” (Takydromus guilinensis up to 36, other species over 36 - T. intermedius, T. yunkaiensis - 47, T. albomaculosus – 52, T. sylvaticus - 67); MBSR - “scales in a transverse row at mid-body, including ventrals” (T. guilinensis up to 33, other species over 40 - T. yunkaiensis and T. intermedius; 42 - T. albomaculosus and 45 - T. sylvaticus).

>> Thanks for your suggestion. We changed the sentences to “Morphologically, T. guilinensis sp. nov. differs from other four Takydromus species from the same clade more evidently in longitudinal rows of dorsal scales (up to 36 in T. guilinensis, but at least 36 in T. intermedius, 47 in T. yunkaiensis, 52 in albomaculosus, and 67 in T. sylvaticus), transverse rows of scales at the mid-body (up to 33 in T. guilinensis, but at least 40 in T. intermedius and T. yunkaiensis, 42 in T. albomaculosus, and 45 in T. sylvaticus), and mensural variables (Table 3; Figure 3)”

Design “Figure 1. A gravid female Takydromus guilinensis sp. nov. (A) and the locality where 10 specimens of the new species were collected (B)” requires improvement in part (B).

>> Thanks for your suggestion. We remade Figure 1B to improve its readability

Add to the discussion the publication “Climate-driven mitochondrial selection in lacertid lizards” (Zhang et al., 2024, doi: 10.1002/ece3.11176).

>> Added. Thanks for your suggestion